# Inclusion of Anti-Tick Vaccines into an Integrated Tick Management Program in Mexico: A Public Policy Challenge

**DOI:** 10.3390/vaccines12040403

**Published:** 2024-04-10

**Authors:** Rodrigo Rosario-Cruz, Delia Inés Domínguez-García, Consuelo Almazán

**Affiliations:** 1Biotechnology in Health and Environmental Sciences Research Laboratory, Natural Sciences College, Autónomous Guerrero State University, Chilpancingo 39105, Guerrero, Mexico; 14970@uagro.mx; 2Immunology and Vaccines Laboratory, College of Natural Sciences, Autonomous University of Queretaro, Santiago de Queretaro 76230, Queretaro, Mexico; c_almazan_g@hotmail.com

**Keywords:** ticks, tick vaccines, *Rhipicephalus microplus*, tick control

## Abstract

Acaricides are the most widely used method to control the cattle tick *Rhipicephalus microplus*. However, its use increases production costs, contaminates food and the environment, and directly affects animal and human health. The intensive use of chemical control has resulted in the selection of genes associated with resistance to acaricides, and consumers are increasingly less tolerant of food contamination. This scenario has increased the interest of different research groups around the world for anti-tick vaccine development, in order to reduce the environmental impact, the presence of residues in food, and the harmful effects on animal and human health. There is enough evidence that vaccination with tick antigens induces protection against tick infestations, reducing tick populations and acaricide treatments. Despite the need for an anti-tick vaccine in Mexico, vaccination against ticks has been limited to one vaccine that is used in some regions. The aim of this review is to contribute to the discussion on tick control issues and provide a reference for readers interested in the importance of using anti-tick vaccines encouraging concerted action on the part of Mexican animal health authorities, livestock organizations, cattle producers, and academics. Therefore, it is suggested that an anti-tick vaccine should be included as a part of an integrated tick management program in Mexico.

## 1. Introduction

Arthropods represent 80% of the known species in the animal kingdom and are evolutionarily successful organisms [1], belonging to a group of ectoparasites causing important economic losses in the cattle industry in tropical and subtropical agroecosystems; they are considered the second most important vectors of parasitic diseases worldwide only after the mosquitoes [2,3]. This group of arthropods, is also responsible for more than 100,000 cases of human diseases and are the most important vector of pathogens in wild and domestic animals in North America [4].

Ticks and tick-borne diseases (TTBDs) are a major problem around the world, affecting animal health and food production. The ticks within the group of ectoparasites have adapted to most of the terrestrial niches on the planet and have specialized in blood feeding on mammals, birds, and reptiles [5,6,7]. The evolutionary adaptation of ticks to hematophagy is the main reason for the great economic losses caused by this group of parasites. However, the biggest impacts of ticks on human and animal health are the diseases they transmit.

The cattle tick *Rhipicephalus microplus* causes direct damage due to the action of bites and its hematophagous behavior [8], as well as indirect damage caused by the transmission of pathogens such as *Babesia bovis*, *B. bigemina*, and *Anaplasma marginale* [9]. Prior to the eradication of the cattle ticks *R. microplus* and *R. annulatus* in the US, indirect economic annual losses caused by cattle babesiosis were estimated at USD 130.5 million (currently equivalent to USD 3 billion). It has been calculated that if ticks had not been eradicated from the US, the livestock industry’s annual losses caused by ticks would be approximately USD one billion [10,11].

Recent publications have already discussed the major drawbacks of using acaricides to control ticks. The paradigm shift from the chemical approach to an integrated approach, in order to bring all previous knowledge into a successful One Health collaborative approach, includes research institutions, industry, the federal government, educational groups, and policy makers, in order to provide the basic requirements to transform the idea into a coherent science-based One Health program [12]. The next point is the science-based government policies, derived from a collaborative, surveillance, and vaccination program, to control ticks and tick-borne diseases and mitigate acaricide resistance as well.

The aim of this review is to contribute to the discussion on the *R. microplus* tick control issues as well as to provide a reference for all those interested in the importance of anti-tick vaccine development and application in order to decrease the use of acaricides and consequently mitigate acaricide resistance; it also aims to encourage concerted action on the part of animal health authorities, livestock organizations, cattle producers, and academics to establish integrated programs of tick control including anti-tick vaccines.

## 2. Biological Diversity and Taxonomy of Ticks

Arthropods belong to a large animal phylum of veterinary and medical importance [13,14,15], including ticks as the most important vectors of pathogens causing diseases in wild and domestic animals [2].

The list of tick names contains 908 total valid species. The family Argasidae includes 186 valid species and the Ixodidae 721 species; one single species is included within the Nuttalliellidae family, named *Nuttalliella namaqua* [16]. Evolutionary and tick taxonomic relationships are an active area studied by using a wide variety of molecular tools [17].

Two new cretaceous fossil species were identified and included in the list of valid names. The species *Deinocroton draculi* was included within a new family named Deinocrotonidae and the other fossil species, named *Cornopalpatum burmanicum*, was included within the family ixodidae [18]. Therefore, the taxonomic scenario of ticks currently includes four families: Argasidae (186), Ixodidae (722), Nutalliellidae (1), and Deinocrotonidae (1), totaling 910 valid names including the two new cretaceous fossil species recently discovered and included within the families, Deinocrotonidae (1) and Ixodidae (1), respectively [18].

The life cycle, host interactions, and other aspects of the biology of both argasid and ixodid ticks vary significantly; for instance, the number of instars during their life cycle varies from two to eight instars, but each instar requires a blood meal to progress to the next developmental stage. Other evolutionary life cycle adaptations resulted in different blood-feeding patterns, involving one, two, or three hosts, depending upon the tick species [19,20,21,22].

Ticks are telmophagous organisms and use their cutting mouthparts to lacerate blood vessels, creating feeding “pools” of blood [23]. However, ticks’ mouthparts are different; while argasid chelicerae are more specialized for skin cutting, ixodids have a well-developed hypostome to facilitate long-term host attachment [24]. Many ixodid tick species also produce a substance called cement to help their attachment to the host during the course of feeding that may last for more than a week [25]. Attachment cement deposit patterns differ among ixodid genera according to the mouthpart structure [26].

Other important differences between argasids and ixodids are the frequency and duration of blood feeding as well as the host exposure to tick saliva [24]. Argasid ticks blood feed for up to two hours and can experience a twelve-fold body weight increase [27]; meanwhile, a fully engorged female ixodid tick blood feeds for more than a week and may increase its weight from 100- to 200-fold as much [25,27,28].

The use of chemical compounds with a lethal effect on arthropods is the most commonly used method to combat ticks [29]. Intensive use of chemicals in combination with the plasticity of tick genomes has led to acaricide resistance (Figure 1), and in many cases to the appearance of multiple resistance, leading to a significant environmental impact produced by the contamination of the soil, subsoil, and water, as well as the effect on other beneficial arthropods and the presence of toxic residues in milk, meat, and other subproducts destined for human consumption derived from this type of livestock production system [30]. Acaricide resistance is the major drawback of the use of acaricides (Figure 1). The selection of resistant tick populations is due to the ignorance of the mechanisms of action and the overuse of mixtures of acaricides prepared based on wrong concepts [31]. As a consequence, the half-life of chemicals used in some regions of northern Mexico has been reduced to such a level that they currently do not represent an alternative to control ticks.

## 3. Acaricide Resistance and Food Safety

Food is essential to life; hence, food safety is a basic human right. Billions of people worldwide are at risk of unsafe food or of becoming sick while hundreds of thousands die yearly from its consumption [32].

Food safety is currently one of the most important problems of the 21st century. The demand for food safety and attempts to minimize environmental impact are two ideas that have influenced the need for a change in pest management strategies. This paradigm shift is rapidly moving from chemical control strategies towards sustainable technologies in order to mitigate the environmental effects of pesticides, considering that the distribution and spread of pests are determined by biological, environmental, and economic factors, such as global warming and economic globalization, respectively.

Acaricides have played an important role in food production and the economic reduction of losses caused by pests, and most likely, they will continue to be important in protecting livestock or agricultural practices from pests.

Furthermore, the use of highly toxic systemic acaricides that are used to control cattle ticks contaminate food derived from the livestock or agricultural industry, such as meat, milk, or agrifood products, becoming a serious health problem for consumers.

The continued use of acaricides and the appearance of acaricide resistance are two linked events in a kind of “microevolution”, in which the chemical treatment acts as the selection pressure, and the resistance as the selected characteristic that confers to the organisms the ability to survive a toxic environment, making the control of parasites more difficult (Figure 1) [33]. In addition, there are pesticides such as macrocyclic lactones that are excreted through urine, milk, and feces; for this reason, anti-tick vaccines promise to be a viable alternative to improve animal health, reduce environmental contamination, and contribute to the production of safe food derived from livestock and agricultural activity [34].

## 4. Ticks and Tick-Borne Pathogens: A Public Policy Issue

Ticks transmit a wide range of pathogens affecting human and animal health, the biology of ticks is connected to environmental and animal host factors, and TBDs can have a significant impact on public health and the livestock industry. The One Health concept recognizes the interdependence between human, animal, and environmental health (Figure 2). It also emphasizes the importance of collaboration and communication among disciplines, including medicine, veterinary medicine, public health, environmental health, and others, to achieve optimal health outcomes for the whole ecosystem [35,36]. In other words, One Health recognizes that humans, animals, and the environment are all interconnected and that the health of one depends on the health of the others (Figure 2).

One Health seeks to promote collaboration and cooperation across different expertise fields, in order to address health challenges and promote health and well-being for all stakeholders.

Increasing numbers of new cases of TBDs and populations of medically important ticks have been identified in recent decades, occupying expanding geographic areas; in addition, an increasing number of human pathogens such as tick-borne bacteria, viruses, and protozoa have been recognized to contribute to the increasing number of TBDs in Mexico and the United States [37]. As a result, the prevention and diagnosis of TBDs are becoming a mandatory health issue, but it greatly depends on the accurate understanding of their epidemiology and distribution by the public and health care providers, as well as the accurate localization maps pinpointing where persons are exposed to TTBDs. However, in Mexico, there are several gaps in the distribution of medically important ticks, and the data on the prevalence of TBDs are still incomplete or outdated. Therefore, efforts to accurately depict the geographical risks are hampered by the lack of systematic surveillance for medically important TTBDs. Control of TTBDs requires an interdisciplinary collaborative effort to approach such health emergencies, and the management of their biological, social, and political components [38,39].

The One Health paradigm has been adopted as an institutional tripartite initiative by the World Health Organization (WHO), the Food and Agriculture Organization of the United Nations (FAO), and the World Organization for Animal Health (WOAH) [36], recognizing the interconnection and interdependence of human, animal, and environmental health within the global security concept of One Health. It also recognizes the importance of dismantling disciplinary groups, replacing them with multisectoral, multidisciplinary, interdisciplinary, and transdisciplinary work teams, and strengthening efforts to detect and respond to threats against human, animal, and environmental health (Figure 2) [36]. A One Health approach is a mandatory initiative for ensuring effective and sustainable efforts to prevent tick infestations as well as TBDs [35].

## 5. The Cattle Tick Eradication Program in Mexico

The cattle tick *R. microplus* is native to Asia, where it evolved as an ectoparasite of Zebu cattle (*Bos indicus* and *Bos Taurus*) and was then dispersed to the rest of the world by commercial cattle transportation [40]; it is also endemic in tropical and subtropical regions around the world [41]. On the American continent, it is found from the Southern US to Argentina, and it is considered to be the most important tick, due to its distribution, capacity to cause damage, and the number of cattle it affects [42].

The cattle tick *R. microplus* is also included in a large group of medical and veterinary importance ticks due to the increasing number of pathogens they transmit and the direct damage caused to the skin of companion, domestic, and wildlife animal species [43,44,45,46,47].

Before the US cattle tick eradication program (CTEP) that started in 1906, the cattle industry was devastated by cattle babesiosis, known as Texas cattle tick fever at that time, caused by two TBPs: *Babesia bovis* and *B. Bigemia*, both transmitted by *R. microplus* and *R. annulatus*. These ticks were eradicated from the US after 40 years of effort and the application of an aggressive eradication program that included the treatment of arsenicals using dipping vats [48]. However, a permanent quarantine line along the Mexican borderline was established to prevent tick reinfestation in 1943 [49].

The Texas Animal Health Commission (TAHC) expanded the preventive quarantine zone in South Texas because of the presence of resistant ticks on livestock and wildlife in 139 grassland areas [50].

Efforts to control cattle ticks in Mexico were then adapted from the successful US CTEP, according to the provisions of the official National Campaign against *R. microplus*, which began with a loan from the Inter-American Development Bank in 1975. The initial credit from the Development Bank ended in 1981 and an additional line of credit was approved; however, the Mexican government decided not to accept the new loan, based on the changing economic and financial situation of Mexico in the 1980s [48], as well as the emerging resistance to coumaphos, the ixodicide officially used during the program [51]. Since then, the principal activities of the National Program of Tick Control in Mexico are the application of tick-killing treatments, the shipment of specimens for taxonomic identification, the diagnosis of acaricide resistance, the epidemiological surveillance of hemoparasitic diseases, attention to production facilities infested with ticks resistant to different acaricides, and the training and dissemination of the Campaign to producers and veterinarians [52].

Currently, more than 52% of Mexican territory is infested with *R*. *microplus* or *R*. *annulatus* [53], and approximately 75% of cattle are at risk of acquiring babesiosis [54,55,56]. However, despite the risk, the importation of live animals from Mexico into the USA represents a lucrative business; back in 2019, 1.3 million live cattle were imported from Mexico [57]. Currently, importation by the USA requires that cattle from Mexico have a tick-free status certification according to the official protocols of both countries [58].

The free trade agreement and the need for tick control in both Mexico and the USA require concerted action between both countries to address issues that could disrupt the binational cattle trade industry, stressing the income it represents for Mexico’s foreign exchange, including annual earnings of USD 700 million [59,60]. This issue can be achieved through the continued exchange of scientific and technological information and its translation into protocols and regulations concerning the livestock industries of both countries [53].

Research coordination on the integrated management of ticks is also required between Mexico and the USA in order to unify binational efforts against ticks involving the Mexican and American CTEPs [55,59,60]. The control of TTBDs is in both countries’ interests and an important animal health issue [61] that requires the participation of stakeholders from Mexico and the USA, livestock producers, the pharmaceutical industry, government regulatory agencies, and research institutions to discuss the research and knowledge gaps that require attention in order to make progress on integrated management strategies for the prevention and control of TTBD infestations and infections, respectively [60], including tick vaccine development supported by both countries and strengthened by the pharmaceutical industry, in order to promote the environmental protection, food safety, and rational use of acaricides and acaricide resistance prevention, ending with science-based feedback from national CTEPs to control TTBDs, which represent a huge challenge (Figure 3).

The control of TTBDs in Latin American countries is a prevailing need due to the economic losses they cause. The use of chemical compounds with a lethal effect on arthropods is the most common available method to combat ticks [28]. Intensive use of chemicals in combination with the plasticity of tick genomes has inevitably led to the emergence of resistance to different families of acaricides and in many cases to the appearance of multiple resistance; to soil, subsoil, and water contamination; to the effects on other beneficial species of arthropods; and to toxic residues in milk, meat, and other products for human consumption derived from this type of livestock production activities (Figure 2) [30]. The emergence of such resistance is the major drawback derived from the use of acaricides, and the selection of resistant tick populations is due to the ignorance of the mechanisms of action and the overuse of mixed acaricides, which is probably the cause of the growing multiple acaricide resistance problem in Mexico over the last 2–3 decades [31]. As a consequence, the half-life of acaricides currently used to control ticks in some regions of northern Mexico has been reduced to such a level that chemical control no longer represents an alternative to control ticks (Figure 1).

## 6. Host Resistance to Ticks and Anti-Tick Vaccines

Cattle immune response to ticks was observed since the beginning of the last century when Australian researchers Johnston and Bancroft (1918) [62] documented that after successive infestations by the cattle tick *R. australis*, cattle became immune to subsequent tick infestations and suggested that cattle developed antibodies against the substances inoculated during tick feeding. However, it was Trager (1939) [63] who described that the immune reaction occurred upon successive infestations, and decreased tick populations in laboratory animals. Interestingly, almost four decades without research on anti-tick vaccines passed and that is explained because synthetic pesticides such as DDT, BHC, aldrin, dieldrin, endrin, chlordane, parathion, etc., were massively produced during the 1940s when these chemicals were highly effective against several arthropod pests [64].

The proof of concept of tick vaccination came out after the experiments performed by Allen and Humphreys (1979) [65], who described the first immunization experimental trial and tick challenge in two separate experiments using guinea pigs and cattle. The authors hypothesized that immunization against the one-host tick *R. microplus* would be more effective than in three-host ticks since antibodies present in blood meals are uptaken during the three developmental stages that see ticks feed on the same immunized animal.

There is a growing interest in the field of vaccinology based on those new concepts called ‘vaccinomics’. The overall idea behind vaccinomics is to understand the mechanisms and pathways that determine immune response to identify new candidate antigens. These new concepts are represented in Figure 4, showing how anti-tick vaccines can be used in order to improve environmental, animal, and human health, derived from the intensive use of acaricides used for tick control.

## 7. Bm86-Based Vaccines

Bm86 is an 89 kDa membrane-bound extracellular glycoprotein (Table 1) identified in *R. microplus* in 1986 [66]. This glycoprotein is located on the surface of the *R. microplus* tick intestinal cells [67]. Further studies by Rand et al. (1989) [68] concluded that the nucleotide sequence of *Bm86* contains a 1982 base pair open reading frame with a prediction of 660 aa including a 19 aa long signal sequence and a 23 aa long hydrophobic region adjacent to the carboxyl terminus.

The identified Bm86 antigen was expressed in *Escherichia coli* and used as a formulation of the first recombinant vaccine produced against an arthropod [68]. The immunization of cattle with Bm86 vaccines produced a 70% to 90% reduction in tick infestation (Table 1) [69], and up to 60% of females and males presented damaged distended intestines with blood leakage into the hemolymph and affected females were unable to engorge or eggs [70].

**Table 1 vaccines-12-00403-t001:** Tick vaccine trials that have been performed in Mexico testing the protective efficacy of several antigens against *R. microplus* infestations on immunized cattle.

Tick Vaccine Antigen	Localization/Function	General Efficacy	References
Bm86/Bm95	Membrane-bound extracellular protein, intestinal cells	49–70%	[71,72,73]
Subolesin	Highly conserved protein involved in tick feeding and fertility	51–60%	[74,75]
Bm95-msp1a	Tick intestine glycoprotein and *Anaplasma marginale* msp1a	64%	[76]
Sub-mp1a	Tick feeding and fertility, and *A. marginale* msp1a	81%	[76]
Ba86	Bm86 orthologue from *R. annulatus*, Mercedes strain	71.5%	[77]
Ferritin 2	Tick iron metabolism, originally identified in *Ixodes ricinus*	64–74%	[78]
Subolesin peptide	Subolesin immunogenic peptide derived from *R. microplus* Media Joya strain	67%	[79]
VDAC	Mitochondrial protein with a role as a central component of the apoptotic machinery in *R. microplus*	82%	[80]
Bm86 polypeptide	Bm86 polypeptide derived from *R. microplus* Media Joya strain	58%	[81]

The most important effects observed from *R. microplus* feeding on vaccinated cattle include a significant reduction in the number of engorged female ticks, their weight, and reproductive capacity, and after several generations, a significant reduction in the tick population [82,83] (Figure 4).

The production and commercialization of the Bm86 under the name TickGard started in Australia in 1994 [82]. However, because of the cost of vaccinating animals three times per year, anti-tick vaccines became unpopular, representing low sales; the production company decided to retire TickGard from the market in 2002 [84].

The Bm86 antigen was also isolated from a Cuban tick strain and expressed in the yeast *Pichia pastoris* by Rodriguez et al. (1994) [83]. Gavac, the commercial vaccine containing *rBm86* started in Latin America in 1995 and is the only anti-tick vaccine currently available in the market. More than 3 million cattle have been vaccinated since its introduction [73].

In Mexico, the introduction of Bm86-based vaccines started in 1997. Anti-tick vaccines were then applied as part of a subsidized program by some state governments, and by the end of the program, only a few producers continued vaccinating cattle pending the availability of tick vaccines. In one farm in Soto la Marina, Tamps., a reduction of 67% in treatments with acaricides was obtained after 10 years of vaccination [85]. However, acaricide treatments were still required to treat infestations by *Amblyomma* spp., the second tick of importance in the region. This is, in fact, one of the main disadvantages of Bm86 vaccines.

The variation in efficacy of Bm86-vaccinated cattle among different geographical isolates was observed particularly in an Argentinian strain where it was found that a 3.4% difference in the amino acid sequence of Bm86 between the tick vaccine strains and the field strain was responsible for a decrease in vaccine efficacy [86]. To address this issue, other antigens homologous of Bm86 have been investigated and resulted in the identification of Bm95, a protein of 75–80 kDa and 569 aa, isolated from a *R. microplus* Argentinian tick strain [87]. This vaccine conferred 89% efficacy against a Cuban tick strain (Camcord) and 58% against an Argentinian strain [87]. These results suggested that a broad-spectrum vaccine could be obtained with the inclusion of antigens from different geographical tick isolates. However, this strategy remains poorly explored; but other strategies like using Bm86 antigenic peptides have been investigated. Recently, a Bm86 polypeptide was designed and tested in a controlled cattle pen trail against *R. microplus*, in the Mexican tropics, obtaining an overall efficacy of 58% [81]. 

## 8. Other Available Proteins

### 8.1. Subolesin

Subolesin (Subn) was identified after several rounds of vaccination with cDNAs from a library of *I. scapularis* embryos [88]. Subn is an orthologue of insect akirin; it is highly conserved among tick species, and it has been involved in gene expression and several cellular pathways (Table 1) [89]. Functional studies using RNAi in ticks concluded that Subn is involved in tick feeding and fertility [90]. Complementary DNA (cDNA) was selected based on RNAi characterization in recently molted adult *R. microplus* tick females and engorged females. These selected cDNAs were tested as recombinant proteins in pen trials. In comparison to other selected cDNAs, the silencing of Subn produced high mortality and low feeding and reproduction in *R. microplus* female ticks. Therefore, a recombinant protein was expressed and evaluated in vaccination trials against *R. microplus* and *R. annulatus* resulting in 51 and 60% efficacy, respectively [91].

### 8.2. Chimeric Proteins

The major surface protein (MSP), described in *A. marginale* from infected bovine erythrocytes, is a heterodimer composed of two structurally unrelated polypeptides: MSP1b and MSP1a. MSP1a is an adhesin for bovine erythrocytes in both native and cultured tick cells [92]. Chimeric proteins comprising the Bm95 or Subolesin immunogenic peptides fused to the *A. marginale* MSP1a N-terminal region (Bm95-MSP1a) (Table 1), exposed on the surface of *E. coli* cell membrane, resulting in a simple and cost-effective process for the production of vaccine preparations involving the propagation and fermentation of the recombinant *E. coli* followed by cell harvesting, disruption, and debris separation. The efficacy in cattle immunized with bacterial fractions containing the chimeric proteins Bm95-MSP1a was 64% [76,92].

The same system was used to produce a chimeric protein formed by Subn antigenic peptides and MSP1a. SUB-MSP1a against *R. microplus* was 81% (Table 1). These results demonstrated that the efficacy against *R. microplus* is improved by using Subn immunogenic peptides [76,92]. More recently, a Subn peptide was tested in vaccine preparations under field conditions showing 67% efficacy against *R. microplus*, confirming the efficacy of Subolesin when using antigenic peptides [79].

## 9. Anti-Tick Vaccination Role as a Part of an Integrated Control Program

Livestock farming in Mexico is an important agricultural activity; it occupies more than 50% of the national territory [93,94]. Dairy farming in Mexico is distributed in different agroecological regions and dairy basins that differ in technology (intensification, production levels, and costs) and depend on the use of specialized dairy breeds (Holstein, Swiss Brown, and Jersey) or crossbred cows (*Bos taurus* x *Bos indicus*). The latter are located in the dual-purpose systems established in the Mexican tropics [95]. Currently, the Mexican inventory is over 36 million heads of cattle; from that, 2,678,557 and 33,661,327 correspond to dairy and beef cattle, respectively. Beef cattle are the most exposed to tick infestations because they are raised in extensive systems. The production of calves resistant to ticks and weather conditions in tropical areas is frequently performed using Zebu cows, in order to procure *Bos taurus* x *Bos indicus* crosses [96] that can be included within an integrated tick control program.

Most of the information on tick vaccines that has been published corresponds to controlled vaccination trials; few evaluations exist under natural conditions, mainly by private companies, but the obtained results are not publicly available.

It is known that anti-tick vaccination is the most important component of an integrated tick control program, including the use of an acaricide. In this vein, there is one study performed in Mexico [97]. The obtained results indicated that when the Bm86 vaccine was used in combination with amidines in a farm with detected resistance to pyrethroids and organophosphates, the efficacy on *R. microplus* control was close to 100% [97].

Although the objective of vaccination is to produce a low level of tick infestations by decreasing the tick progeny, it is also possible to eliminate the *R. annulatus* tick species from a certain region by combining tick vaccines with a long-lasting acaricide [76,98,99]. In 2002, an experimental study published by Arocho et al. in southern Texas [100] demonstrated that vaccination with Bm86 in combination with macrocyclic lactones caused a reduction of the index of fertility for 4.3 and 4 months, respectively, with greater and longer efficacy than separated treatments. 

All the studies shown herein demonstrate the feasibility of controlling ticks through vaccination as a part of an integrated tick control program (ITCP). However, variation between tick strains should be considered [75,76,86]. Also, individual variation in immunized hosts, population dynamics, and the tick biology of local strains require attention.

Immunological studies carried out in the Pacific coastal State of Guerrero, Mexico, using the Bm86 protein obtained from local ticks, have shown that the use of an ITCP, an alternate methodology including a chemical and immunological method to control ticks, produced a significant reduction on the average tick weight and the number of acaricide applications per year [101]. The use of recombinant vaccines represents an alternative in tick control when used under an integrated control scheme in combination with acaricides; however, concerted action from animal health authorities, livestock organizations, and private and public institutions is required to make tick vaccines available to establish integrated tick control programs in Mexico.

While new antigens are identified, and multi-antigenic and multi-epitope vaccines able to reduce tick infestations and block TBPs are developed, the already-identified and tested antigens can help to reduce *R. microplus* infestations in cattle and mitigate acaricide resistance.

## 10. Tick Control and Vaccination Proposal

According to the articles reviewed herein on experimental tick vaccination trials, and a combination of acaricides and vaccination, we propose that tick control programs in Mexico should be established with the participation of (a) official experts from the National Center of Parasitology (SENASICA); (b) the researchers from institutions involved in tick vaccine development; (c) the veterinary pharmaceutical industry; and d) the local livestock and producer associations. In addition to the technical advice and expertise on acaricide resistance provided by SENASICA, the existing and newly generated information on population dynamics, tick biology, and tick resistance should be considered. The acaricides are an essential part of an integrated tick control program and should be proposed based on their efficacy and the susceptibility of local tick strains. The acaricides should be applied by certified veterinarians under strict biosafety conditions, sharing responsibility and collaboration with pharmaceutical companies that provide acaricides. After acaricide treatment, the total amount of cattle exposed to infested pastures in a farm should be vaccinated. Therefore, acaricides will eliminate parasitic tick phases, while antibodies produced after vaccination will protect cattle against tick reinfestations from successive generations. It is expected that a combination of both acaricides and vaccination will reduce the tick burden to a point where ticks are not harmful to cattle, as well as cases of TBD such as babesiosis and anaplasmosis. Anti-tick vaccination in combination with acaricides will positively impact the cattle industry through a reduction in the chemical contamination of milk and meat, a decrease in the frequency and cost of acaricides, and mitigation of tick resistance; however, as it was already mentioned, a joint effort of organized producers, the pharmaceutical industry, and the government official experts from SENASICA is still a mandatory issue.

## 11. Conclusions

Ticks are widely distributed in Mexico; therefore, a continued monitoring and surveillance program, should be a mandatory educational initiative for producers and workers, who should be aware of the risk of cattle contracting TBDs. 

By incorporating the basic concepts of human, animal, and environmental health, as well as a collaborative multi-institutional surveillance consortium, decision makers will be able to establish the basis of a One Health enterprise (Figure 2) that will place the science and management of TBDs in a broader biological and socio-ecological context, in order to build up intelligent public policies for the benefit of producers and stakeholders.

Integrated tick control management should be considered an important instrument of public policies to control TTBDs, since it only requires a combination of a few environmentally friendly control methods including cattle management, the rational use of acaricides, and an anti-tick vaccine, in order to reduce the use of chemicals, prevent environmental contamination, and mitigate acaricide resistance.

Vaccines should be included in official tick control programs because they reduce the use of acaricides affecting both animal and human health. However, today, tick control is still a great challenge that needs the joint effort of scientists, livestock producers, and decision makers in Mexico and around the world.

## Figures and Tables

**Figure 1 vaccines-12-00403-f001:**
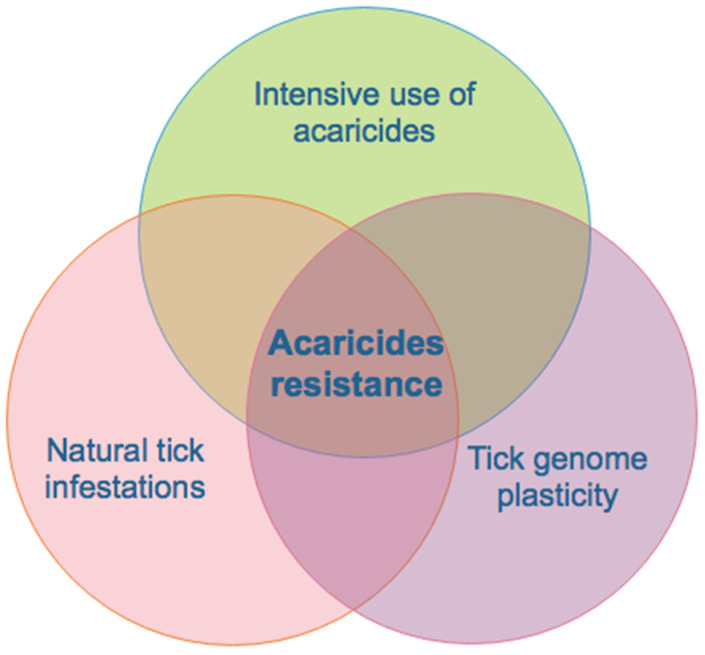
The frequent use of acaricides to control ticks acts as a selection pressure directly on their genome, which implies the selection of genes associated with resistance to acaricides in the progeny of ticks. This process will invariably lead to the gradual emergence of resistance to acaricides in the short term. (Artwork by Fernando Rosario Dominguez).

**Figure 2 vaccines-12-00403-f002:**
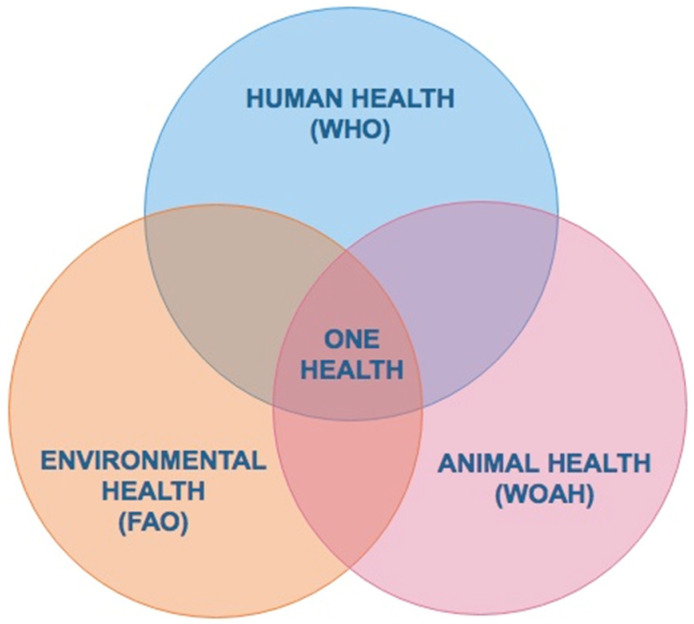
The One Health paradigm is a tripartite initiative by the World Health Organization (WHO), the Food and Agriculture Organization of the United Nations (FAO), and the World Organization for Animal Health (WOAH), recognizing the interconnection and interdependence of human, animal, and environmental health, within the concept of one single health. (Artwork by Fernando Rosario Dominguez).

**Figure 3 vaccines-12-00403-f003:**
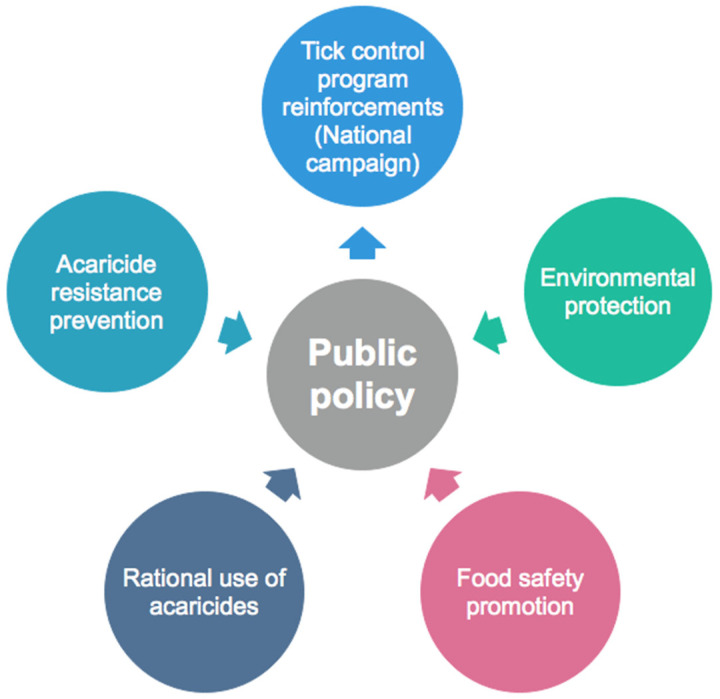
Cattle tick control is an issue that must be addressed in an integrated manner, so that public policies will have a scientific basis, leading to intelligent solutions and research actions focused on the control of TTBDs and the promotion of animal health, food safety, and environmental protection. (Artwork by Fernando Rosario Dominguez).

**Figure 4 vaccines-12-00403-f004:**
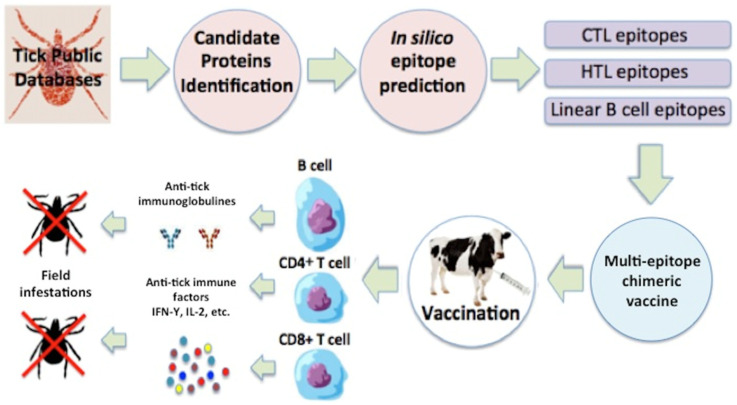
Reverse vaccinology has evolved from the use of single-gene recombinant vaccines to multi-epitope chimeric vaccines (a vaccinomic approach); however, the vaccine’s mode of action to control ticks after tick feeding remains the same in both cases. CTL, cytotoxic T lymphocytes; HTL, helper T lymphocytes; IFN, interferon; IL, interleukin. (Artwork by Fernando Rosario Dominguez).

## Data Availability

The data shown in this study are available in this publication.

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
