# Peer review of "Inclusion of Anti-Tick Vaccines into an Integrated Tick Management Program in Mexico: A Public Policy Challenge"

_vaccines, 2024, doi:10.3390/vaccines12040403_

Round 1

Reviewer 1 Report

Comments and Suggestions for Authors

This paper jumps around a lot. There are a lot of sloppy formatting errors such as capitals in some headings and not others. A greater in depth review on the state of anti-tick vaccination in Mexico was indicated in the abstract but not discussed in the paper. This would have been more beneficial. Section relating to the evolutionary adaptation in ticks should not be present and was redundant. As this is a veterinary tick, more focus should be placed on this aspect and less on the one health aspect. A lot of topics were mentioned and glossed over which defeats the point of a review. I suggest limiting the scope and going into more detail on specific aspects relevant to this tick species in this geographic location. 

Comments on the Quality of English Language

The english is very poor in this manuscript. Rhipicephalus microplus is in full where it should be abbreviated and abbreviated where it should be in full. No consistent use of scientific names throughout.  Numerous grammatical errors found.

Author Response

We really appreciate your comments, they have been very helpfull.

Rodrigo Rosario (on behalf of the co-authors).

Reviewer 2 Report

Comments and Suggestions for Authors

This review of tick management offers the readers a good background in the history of tick management, along with suggestions that adding a tick vaccine to the management might further limit tick infestations and disease. As acaricides are apparently continue to be needed if anti-tick vaccines are employed, the emphasis on their lack of utility is confusing, and needs to be better addressed. Also, there needs to be better evidence that anti-tick vaccines will be helpful; and some revision in the text is needed to reflect more caution in that regard.

Comments on the Quality of English Language

Some grammar needs to be corrected.

Author Response

We greatly appreciate your effort to review this document, your comments were a big help to the improvement of the manuscript.

Sincerely

Rodrigo (On behalf of the co-authors)

Reviewer 3 Report

Comments and Suggestions for Authors

The paper is a review/proposal on inclusion of anti-tick vaccine in Mexico. It is in the similar line as a paper published in Vaccines which was fairly complete wih 238 references  :van Oosterwijk JG, Wikel SK. Resistance to Ticks and the Path to Anti-Tick and Transmission Blocking Vaccines. Vaccines (Basel). 2021 Jul 2;9(7):725. doi: 10.3390/vaccines9070725.

The autors  should come more rapidly to the Mexico situation due to the existence of this former general paper.

The eradication is taken as possibility, a too simple possibility.

The USDA implemented the tick eradication program, at great scale and expense since 1906 concerning Rhipicephalus (Boophilus) microplus. The cattle tick fever across the US was eradicated. It is today confined to the Permanent Tick Quarantine Zone between the Texas towns of Del Rio and Brownsville. There are still outbreaks and they should be tracked. Eradication is still challenging although much has been spent .Is Mexico ready for such a long adventure? It will be tough since vaccination has an efficacy at best of 60% and the resistance to drugs largely distributed. I am surprised that selection for resistant animals has not been considered see:

Shyma, K.P., Gupta, J.P. & Singh, V. Breeding strategies for tick resistance in tropical cattle: a sustainable approach for tick control. J Parasit Dis 39, 1–6 (2015). https://doi.org/10.1007/s12639-013-0294-5

Hewetson, R.W., 1968. Resistance of cattle to tick, Boophilus microplus. II. The inheritance of resistance to experimental infestations. Aust. J. Agric. Res. 19, 497-505.

Budeli MA, KA Nephawe, D Norris, NW Selapa, L Bergh, A MaiwasheGenetic parameter estimates for tick resistance in Bonsmara cattle. South African Journal of Animal Science, 2009, 39, 321-327.

I understand that selection for resistance to ticks is a long process. The heritabilities are reasonable (0.3 ?)  The"eradication" is a long process as well and Ido not see why selection could not be included; either the standard way or possibly using genetical markers.

The manuscript should be revised ; 1) concentrate on Mexico situation since a general review exists, 2) try to foresee the feasability of eradication in a realistic way in the situation of the country 3) include or exclude the opportunity of selecting resistant cattle.

Author Response

I really appreciate your comments on the manuscript, I hope you read soon the new version, It will be better.

Thanks

Rodrigo

Reviewer 4 Report

Comments and Suggestions for Authors

Dear authors,

In your review, the importance of vaccination against ticks has been discussed with the Mexican example, and important information has been brought together in the light of current literature on the subject.

I hope that my suggestions will contribute to a better presentation of the subject.

Best regards…

Abstract:

The abstract is expected to better reflect the content of the compilation. The compilation is based on the Rhipicephalus microplus challenge, and the word R. microplus is also used in the keywords, but there is no statement about it in the abstract. The abstract should be edited to include the R. microplus, Mexica sample and its consequences.

Please delete the sentences; ..Cattle ticks affects livestock worldwide, causing economic losses calculated in 20 billions USD per year globall..

Content:

12 subtitles are listed in this section and are suitable for the content of the study.

Introduction:

1.      Please delete …reservoirs…. (line 44), and combine the first and second paragraphs to give the vectoring of ticks in a single paragraph.

2.      Please Delete line 55-56 (However……….) and add…..The biggest impact of ticks on humans and animals is the diseases they transmit.

3.      Please write the species names clearly when they are first written (Line 57. Rhipicephalus microplus)

Evolutionary adaptations of Ticks:

1.      Under this subtitle, the biological differences between argasid and ixodid ticks are given very superficially and the content does not reflect this title.

2.      Line 70-71:  “Arthropod-borne pathogens account for more than 20% out of the total cases of emerging infectious diseases reports documented between 1940 and 2004”

Sentences explaining the importance of such ticks were given in the introduction. Please not to repeat. Add this kind sentences there.

Acaricides resistance, and food safety:

Under this subtitle, it is explained in very general terms that acaridites cause residues. It would be appropriate to give examples in the light of the literature, which acaricides cause residues in which foods.

Ticks and Tick-Borne Pathogens: A One-Health Approach:

This subtitle is a very strong statement. Beyond reviews, books have been written on this subject and even scientific organizations have been organized. It will be very difficult to fill its content. There are nice expressions in this section of the presented manuscript, but it is expected to contain disease examples that will establish the tickk-borne relationship.

The Cattle tick Eradication Program in Mexico

Very important information is given in this subtitle. Could you please give the Mexican example, how it was applied, the results and the points that need to be taken into consideration, in bullet points or as a table, it is a very important subject and will be very useful to the readers.

Line 176: Please do not abbreviate species names at the beginning of sentences.

Line 179-184: There is a problem with the font or size.

Line 181: Please write the species names harmoniously in the text (R. (Boophilus) microplus and R (B) annulatus.)

Line 188: Rhipicephalus (Boophilus) microplus,…..

Line 224 … need due to the million-dollar economic losses they cause, line 203: hat could disrupt the million dollar binational cattle trade industry (Please look at indroduction, there is thise kin of sentence?)

Host-resistance to ticks and anti-tick vaccines:

Figure 4 should be explained in this section.

Bm86-based vaccines (Please add 7):

Enough information has been evaluated under this subtitle.

Other available proteins:

Enough information has been evaluated under this subtitle.

Anti-Tick vaccination role as a part of an integrated control program:

Line 341-342 : ….All the previous studies shown herein demonstrate the feasibility of control ticks through vaccination as part of an 342 integrated tick control program (ITCP). ….Which studies?

Lİne 345-349. Need to refrences.

Tick control and vaccination proposal

Conclusions

The results contain general classical information, It has been rewritten to include results based on the Mexican example, which is highlighted in the review.

Line 371: ….As human cases of tick-borne disease continue to rise in Mexico…. We do not have the information (literatures) to reach this conclusion in the review.

Line 380: …..Vaccines should be included in official tick control programs … You are right, but as we emphasized in the review, we have a very limited vaccine.

References

While making the corrections, references should be rearranged.

Author Response

The authors appreciate your time and effort to review this paper, it was a really helpfull list of corrections and suggestions that really improved the quality of the .

Thanks

Rodrigo Rosario

Round 2

Reviewer 3 Report

Comments and Suggestions for Authors

A paper has similarities with the present paper as I indicated in the first review. It was published in Vaccines which was fairly complete wih 238 references : van Oosterwijk JG,

Wikel SK. Resistance to Ticks and the Path to Anti-Tick and Transmission Blocking

Vaccines. Vaccines (Basel). 2021 Jul 2;9(7):725. doi: 10.3390/vaccines9070725. It should be included in the references. It should be presented in the introduction and then explained what present paper adds to this former recent one in Vaccines.

One other comment was “The USDA implemented the tick eradication program, at great scale and expense since 1906 concerning Rhipicephalus (Boophilus) microplus. The cattle tick fever across the US was eradicated. It is today confined to the Permanent Tick Quarantine Zone between the Texas towns of Del Rio and Brownsville. There are still outbreaks and they should be tracked. Eradication is still challenging although much has been spent. Is Mexico ready for such a long adventure? It will be

tough since vaccination has an efficacy at best of 60% and the resistance to drugs largely

distributed. I am surprised that selection for resistant animals has not been considered ” The authors answered that the farmers are not interested in Bos indicus and prefer imported selected Bos taurus with better production traits. It should then be mentioned from the start that Imported breeds of Bos taurus are the one concerned. Are they the numerically most important? In milk and/or meat productions? It should be clearly stated that selection for resistance is not considered and why in Mexico. Another question was about the economic investment that the state is ready to finance, in comparison with the US eradication. The authors answered that there was also a cultural issue among the farmers in relation to the use of anti-tick vaccine since the ticks did not fall although 60% died. Nothing is discussed in the paper on the efficiency belief of the farmers and how to convince them. This is the core of the use of vaccine when the private farmers have the decision in their hands.

The present paper does not focus enough on the particularities of Mexican situation. There is a small chapter on the diversity of ticks, which is quite general. How Mexico has particular ticks or not?  A specific comment Mexican vaccination was added (lines 411 to 423) which is a good thing. However, there seems to be a repeated sentence on pesticide applications per year (l 418-419), check.  THE addition of a general paragraph on vaccinomics is very general and does not provide information really relevant to the Mexican situation and could be deleted (line 295-302).

There is no evaluation of resistance of ticks to drugs in Mexico. This a difficulty because the association of a vaccine 60% effective with macrocyclic lactones seems to have synergistic effect and is proposed (l 444). The success then relies much on the level of tick resistance to drugs.

I wonder if the general comments on One health in figure 2 are needed since they are too general for this type of paper.

I suggest again to concentrate more on Mexican situation, with data on tick resistance to drugs, farmers reluctance for vaccination, and on which types of cattle are present and how they are concerned with this proposal of eradication.

Comments on the Quality of English Language

Quite easy to read, nothing major.

Author Response

Dear Reviewer, We greatly appreciate your comments, corrections and suggestions, all of them are being a great contribution for the improvement of this manuscript, and we hope that our responses satisfy each comments, if not please do not hesitate to let us know.

1) A paper has similarities with the present paper as I indicated in the first review. It was published in Vaccines which was fairly complete wih 238 references : van Oosterwijk JG, Wikel SK. Resistance to Ticks and the Path to Anti-Tick and Transmission Blocking Vaccines. Vaccines (Basel). 2021 Jul 2;9(7):725. doi: 10.3390/vaccines9070725. It should be included in the references. It should be presented in the introduction and then explained what present paper adds to this former recent one in Vaccines.

We do appreciate your comments.

RESPONSE: We agree, and we have added the suggested reference (doi: 10.3390/vaccines9070725) within the list of the manuscript references (reference included 12) (all references order was corrected). We have also included within the introduction chapter, the next paragraph as suggested:

“Recent publications have already discussed the major drawbacks of using acaricides to control ticks, order to change the paradigm from the chemical to an integrated approach, in order to bring all previous knowledge, into a succesful one health collaborative approach, including research institutions, industry, federal government, educational groups and policy makers, in order to provide the basic requirements to transform the idea into a coherent science based one health program, such as: surveillance strategies, research funding, and public education (reference included 12), before we get to the next point, the science based government policies, derived from the collaborative, surveillance and vaccination program, to control ticks, tick-borne diseases and mitigates acaricide resistance as well.”

This paragraph, emphasize our main point: The huge problem of acaricide resistance in Mexico, require the use of an anti-tick vaccine.

2) One other comment was “The USDA implemented the tick eradication program, at great scale and expense since 1906 concerning Rhipicephalus (Boophilus) microplus. The cattle tick fever across the US was eradicated. It is today confined to the Permanent Tick Quarantine Zone between the Texas towns of Del Rio and Brownsville. There are still outbreaks and they should be tracked. Eradication is still challenging although much has been spent. Is Mexico ready for such a long adventure? It will be

tough since vaccination has an efficacy at best of 60% and the resistance to drugs largely distributed.

Thank you for your comment.

RESPONSE: In the first paragraph you refer to the tick eradication program in the US.

Regarding to the question: Is Mexico ready for such a long adventure? It will be tough since vaccination has an efficacy at best of 60% and the resistance to drugs largely distributed here is our response:

As you mention, the CTEP in the US started at the beginning of the last century. Firstable, to achieve the tick eradication, an intensive use of acaricides was implemented.  By then, secondary effects of those acaricides such as chlorides, DDT, etc. were unknown. Secondly, to avoid dispersion of cattle ticks, thousands of white-tailed were eliminated. Such strategy has ecological implications and cannot be applied nowadays.  Finally, one thing that contributed to the success of the campaign was the weather in the US, where tropical areas are reduced to parts of Southern states of Florida and Texas. In México, by the other hand, in around 60 % of the territory the climate conditions are suitable for Rhipicephalus microplus development. In this territory livestock is one of the most important activity, where the acaricides have been intensively used during decades and tick resistance to most of these products has been reported. Then, we are proposing the anti-tick vaccines as a way to reduce tick populations and mitigate the acaricide resistance, but never to eradicate ticks, that is a big difference between the US approaches currently used to eradicate ticks from Texas, which has been reinfested. Eradication or control?, that is the question. Our proposal has been constructed thinking on tick control.

3) I am surprised that selection for resistant animals has not been considered ” The authors answered that the farmers are not interested in Bos indicus and prefer imported selected Bos taurus with better production traits. It should then be mentioned from the start that Imported breeds of Bos taurus are the one concerned. Are they the numerically most important? In milk and/or meat productions? It should be clearly stated that selection for resistance is not considered and why in Mexico. Another question was about the economic investment that the state is ready to finance, in comparison with the US eradication. The authors answered that there was also a cultural issue among the farmers in relation to the use of anti-tick vaccine since the ticks did not fall although 60% died. Nothing is discussed in the paper on the efficiency belief of the farmers and how to convince them. This is the core of the use of vaccine when the private farmers have the decision in their hands.

RESPONSE: Regarding to your comment on tick-resistant cattle. It is known that Bos indicus are more resistant to ticks, however, due to that Bos taurus are more productive, and because of importation of cattle to the US, cattle producers prefer to raise European breeds.

Are they numerically most important?

RESPONSE: YES

In milk and/or meat productions?

RESPONSE: IN BOTH

Another question was about the economic investment that the state is ready to finance, in comparison with the US eradication.

RESPONSE: The budget from the Mexican Government is applied to the official campaing already mentioned on this manuscript:

The principal activities of the National Program of Tick Control in Mexico, are: application of tick-killing treatments, shipment of specimens for taxonomic identification, diagnosis of acaricide resistance, epidemiological surveillance of hemoparasitic diseases and attention to production facilities, infested with ticks resistant to different ixodicides, as well as training and dissemination of the Campaign to producers and veterinarians (Garrapata Boophilus  spp. Campañas Zoosanitarias, (taken from the referenced Website of SENASICA).  

Nothing is discussed in the paper on the efficiency belief of the farmers and how to convince them. This is the core of the use of vaccine when the private farmers have the decision in their hands.

RESPONSE: This educational program is also included within the official activities of SENASICA, not only for producers but also for Veterinarians to diseminate the campaing as well as the advice on tick control.

We included within chapter 9, the following explanatory paragraph, as suggested.

Livestock farming in Mexico is an important agricultural activity; It occupies more than 50% of the national territory and maintains nearly 32 million heads of cattle. (SAGARPA, 2004a; 2004b). Dairy farming in Mexico is distributed in different agroecological regions and dairy basins that differ in technology (intensification, production levels and costs) and these depend on the use of specialized dairy breeds (Holstein, Swiss Brown and Jersey) or crossbred cows (Bos taurus x Bos indicus). The latter are located in the dual-purpose systems established in the Mexican tropics (Magaña et al, 2003). Currently, the Mexican inventory is over 36 million head of cattle, from that, 2,678,557 and 33,661,327 correspond to dairy and beef cattle respectively. Beef cattle are the most exposed to tick infestations, because they are raised in extensive systems. The production of calves resistant to ticks and weather conditions, in tropical areas is frequently done using Zebu cows, in order to get Bos taurus x Bos indicus crosses (Gonzalez Padilla, et al, 2018), and can be included within an integrated tick control program”.

New references included.

96) SAGARPA, 2004a. Situación actual de la producción de leche en México, 2004. http://www.sagarpa.gob.mx/Dgg.

97) SAGARPA, 2004b. Situación actual y perspectiva de la producción de carne en México, 2004. http://www.sagarpa.gob.mx/Dgg.

98) J. G. Magaña Monforte, G. Ríos Arjona y J. C. Martínez González. Dual purpose cattle production systems and the challenges of the tropics of Mexico. Arch. latinoam. prod. anim. 2006; 14(3); 105-114.

99) Everardo González Padilla, Roger Iván Rodríguez Vivas, Delia Inés Domínguez García, Rodrigo Rosario Cruz, et al  (otros 86 autores). Estado del arte sobre Investigación e innovación Tecnológica en Ganadería bovina tropical. CONACyT-UNAM. 2ª. Ed. (2018). ISBN: 978-607-37-0556-1. pp130-345.

4) The present paper does not focus enough on the particularities of Mexican situation. There is a small chapter on the diversity of ticks, which is quite general. How Mexico has particular ticks or not? 

Thanks for this comment, we are not completely focused on Mexican situation, but on the R. microplus problem in Mexico, certainly the ticks problem in Mexico is much bigger if we take the whole picture of ticks and tick-borne pathogens, but we would like to focuse on the acaricide resistance problem and a proposal to mitigate their effects on the cattle and on the environment; as well as the way it can be approached.

5) A specific comment Mexican vaccination was added (lines 411 to 423) which is a good thing. However, there seems to be a repeated sentence on pesticide applications per year (l 418-419), check.  

Response: Thanks for the correction, we agree, the repetitive sentence has been deleted from the paragraph: “with a similar reduction of aplications (from 15 to 3 aplications/year)” (Deleted sentence).

“These data can be translated in terms of the tick control within the ranch, as a reduction from 15 to 3 aplications of acaricides by the year.  with a similar reduction of aplications (from 15 to 3 aplications/year)” (Deleted sentence).

6) THE addition of a general paragraph on vaccinomics is very general and does not provide information really relevant to the Mexican situation and could be deleted (line 295-302).

Response: Thanks for the suggestion; the authors think that this piece of info contributes to understand the context of tick vaccination.

7) There is no evaluation of resistance of ticks to drugs in Mexico. This a difficulty because the association of a vaccine 60% effective with macrocyclic lactones seems to have synergistic effect and is proposed (l 444). The success then relies much on the level of tick resistance to drugs.

RESPONSE: Must be a missunderstanding, since “evaluation of resistance of ticks to drugs”, is a strong official program in Mexico as we mentioned in a previous chapter of this manuscript, “The principal activities of the National Program of Tick Control in Mexico, are: application of tick-killing treatments, shipment of specimens for taxonomic identification, diagnosis of acaricide resistance, epidemiological surveillance of hemoparasitic diseases and attention to production facilities, infested with ticks resistant to different ixodicides, as well as training and dissemination of the Campaign to producers and veterinarians (Garrapata Boophilus  spp. Campañas Zoosanitarias, (taken from the referenced Website of SENASICA). On the other hand, we do not see a difficulty in using a 60% vaccine and a long lasting acaricide, that program was positively used in Puerto Rico by Arocho back in 2022 (reference 99)”.

8) I wonder if the general comments on One health in figure 2 are needed since they are too general for this type of paper.

RESPONSE: Thanks again. We agree with you, however we decided to keep it there, since it strengthens the one health proposal to control TTBD´s.

9) I suggest again to concentrate more on Mexican situation, with data on tick resistance to drugs, farmers reluctance for vaccination, and on which types of cattle are present and how they are concerned with this proposal of eradication.

Dear reviewer, thank you for your suggestion.

RESPONSE: We decided do not include an extense review on tick resistance because several publications on this issue have been published. Back in 2009 our group published some official info about the multiple acaricide resistance problem in Mexico.

However, a paragraph with a brief summary has been added to the manuscript, in order to make clear the current situation of acaricide resistance in Mexico, “the overuse of mixed acaricides, wich is probably the cause of the growing multiple acaricide resistance in Mexico, over the last 2-3 decades [reference 63]. As a consequence, the half-life of acaricides currently used to control ticks in some regions of northern Mexico, has been reduced to such a level that chemical control, no longer represent an alternative to control ticks (Figure 1)”.

Reviewer 4 Report

Comments and Suggestions for Authors

Dear authors,

Thank you for the revisions. I think this version is better than the previous one.

best regards..

Author Response

Dear Reviewer, We greatly appreciate your comments, corrections and suggestions, all of them are being a great contribution for the improvement of this manuscript, and we hope that our responses satisfy each comments, if not please do not hesitate to let us know.

Round 3

Reviewer 3 Report

Comments and Suggestions for Authors

The paper is OK for me now. Some minor points:

L 46: ,In the US:  write, in the US

L 109: chemicales: write chemicals

L 140: pesticides is rather used for crops. Macrocyclic lactones are anthelmintic and insecticide/acaricide drugs

Author Response

Reviewer 3:

Thank you very much for the corrections made on this manuscript, we greatly appreciate your very helpful comments.

Sincerely

On behlf of my coauthors

Dr. Rodrigo Rosario Cruz

Comments and Suggestions for Authors

The paper is OK for me now. Some minor points:

L 46: ,In the US:  write, in the US

RESPONSE: Correction made

L 109: chemicales: write chemicals

RESPONSE: Correction made

L 140: pesticides is rather used for crops. Macrocyclic lactones are anthelmintic and insecticide/acaricide drugs

RESPONSE: Correction made, we substituted pesticides for: acaricides
